# MSIMG: A Density-Aware Multi-Channel Image Representation Method for Mass Spectrometry

**DOI:** 10.3390/s25206363

**Published:** 2025-10-15

**Authors:** Fengyi Zhang, Boyong Gao, Yinchu Wang, Lin Guo, Wei Zhang, Xingchuang Xiong

**Affiliations:** 1College of Information Engineering, China Jiliang University, Hangzhou 310018, China; fengyi.zhang@cjlu.edu.cn (F.Z.);; 2National Institute of Metrology, Beijing 100029, China; 3Key Laboratory of Metrology Digitalization and Digital Metrology for State Market Regulation, State Administration for Market Regulation, Beijing 100029, China; 4National Metrology Data Center, Beijing 100029, China

**Keywords:** mass spectrometry, density-aware, multi-channel image representation, deep learning

## Abstract

Extracting key features for phenotype classification from high-dimensional and complex mass spectrometry (MS) data presents a significant challenge. Conventional data representation methods, such as traditional peak lists or grid-based imaging strategies, are often hampered by information loss and compromised signal integrity, thereby limiting the performance of downstream deep learning models. To address this issue, we propose a novel data representation framework named MSIMG. Inspired by object detection in computer vision, MSIMG introduces a data-driven, “density-peak-centric” patch selection strategy. This strategy employs density map estimation and non-maximum suppression algorithms to locate the centers of signal-dense regions, which serve as anchors for dynamic, content-aware patch extraction. This process transforms raw mass spectrometry data into a multi-channel image representation with higher information fidelity. Extensive experiments conducted on two public clinical mass spectrometry datasets demonstrate that MSIMG significantly outperforms both the traditional peak list method and the grid-based MetImage approach. This study confirms that the MSIMG framework, through its content-aware patch selection, provides a more information-dense and discriminative data representation paradigm for deep learning models. Our findings highlight the decisive impact of data representation on model performance and successfully demonstrate the immense potential of applying computer vision strategies to analytical chemistry data, paving the way for the development of more robust and precise clinical diagnostic models.

## 1. Introduction

Mass Spectrometry (MS), a core tool in analytical chemistry, plays a pivotal role in frontier life science fields such as metabolomics [1,2,3,4], proteomics [5,6,7], and disease diagnostics [8,9,10]. This technology detects thousands of molecules in biological samples with high sensitivity and resolution, resulting in MS data characterized by high dimensionality, complexity, and large volume [11]. A significant computational challenge in the field is how to efficiently and accurately extract key features associated with specific biological states, such as disease, from this massive and intricate dataset [12]. Consequently, the development of novel and effective computational methods for analyzing MS data is vital not only for elucidating complex biological mechanisms but also for facilitating the early diagnosis of diseases and enabling personalized medicine.

Traditional MS data analysis workflows typically rely on a series of steps including peak detection, metabolite identification, and statistical analysis [13], but this process faces numerous inherent difficulties. For instance, incomplete metabolite annotations lead to the neglect of a large number of signals, algorithmic biases in peak detection and alignment can introduce errors, and unavoidable signal shifts during data acquisition severely affect research reproducibility. Concurrently, artificial intelligence technologies, particularly deep learning [14], have made groundbreaking advancements in fields like computer vision. Models such as Convolutional Neural Networks (CNNs) have demonstrated superhuman performance in tasks like image recognition [15]. Their powerful feature extraction and pattern recognition capabilities offer new approaches for solving complex data analysis problems in other domains. Therefore, integrating MS data with computer vision techniques represents a promising interdisciplinary research direction [16,17].

Fundamentally, MS data can be viewed as a two-dimensional data matrix composed of retention time (RT), mass-to-charge ratio (*m*/*z*), and ion intensity, allowing it to be indirectly treated as an “image” [18]. Utilizing the entire mass spectrum as an “image” for data analysis is an alternative and promising strategy. However, a key engineering challenge is that the dimensions of raw MS matrices are typically enormous. Directly inputting them into deep learning models designed for standard image sizes would lead to an explosion of network parameters. This surge in complexity not only causes catastrophic computational resource consumption but also leads to severe model overfitting. In this context, overfitting means the model memorizes the training data’s noise and specific details instead of learning the core, generalizable patterns. This prevents the model from generalizing to new data, thus limiting its practical utility. Consequently, effective dimensionality reduction or feature selection is imperative before applying deep learning models. To address this challenge, researchers have undertaken a series of pioneering explorations. Shen et al. [18] proposed the deepPseudoMSI method, which converts LC-MS metabolomics data into a fixed-size (e.g., 224 × 224) “pseudo-mass spectrometry image” through a data binning strategy. Specifically, the retention time and *m*/*z* dimensions are divided into a grid, and the intensities of all data points falling within the same grid cell are aggregated to represent the “pixel” intensity. Similarly, Zhang et al. [19] introduced the DIAT data format in proteomics, compressing data by pooling the *m*/*z* dimension and successfully applying it to deep learning models for phenotype classification. While these methods resolve model compatibility issues, their data compression strategies lead to the loss of original mass spectral features. To perform effective feature selection while preserving data characteristics, Wang et al. [20] developed the MetImage method. This approach divides the complete MS matrix into multiple smaller patches using a grid, then selects the most information-rich patches based on entropy or mean values to construct a multi-channel image for input into a deep learning model. This encoding strategy preserves the mass spectral features of the original LC-MS data, enabling the straightforward identification of characteristic metabolites within the key images. This, in turn, helps to elucidate the model’s diagnostic principles, thereby enhancing its interpretability.

Building upon the MetImage framework, this paper proposes a framework for constructing multi-channel mass spectrometry images, named MSIMG. The theoretical foundation of this work is that effective texture feature extraction is critical to the success of texture classification methods [21]. We argue that MetImage has a fundamental flaw in this regard: its fixed-grid partitioning strategy can disrupt the inherent and meaningful texture structures within the data [22], such as splitting a complete signal peak across two or more adjacent patches, thereby compromising feature integrity [23]. To overcome this limitation, MSIMG introduces a novel “density-peak-centric” patch extraction strategy. This approach, inspired by object detection in computer vision, uses Density Map Estimation [24] and Non-Maximum Suppression (NMS) [25] techniques to precisely locate the centers of signal peaks. These centers then serve as anchors for dynamic, content-aware patch extraction, replacing the fixed-grid partitioning of MetImage that can compromise feature integrity. We hypothesize that this “density-peak-centric” patch extraction strategy can more robustly capture the core features associated with biological phenotypes. This study aims to further explore the feasibility, effectiveness, and potential boundaries of applying computer vision techniques to mass spectrometry data analysis through the MSIMG method.

## 2. Materials and Methods

This study proposes a framework for constructing multi-channel image representations of mass spectrometry data, named MSIMG, designed to convert raw LC-MS data into a multi-channel image format suitable for downstream deep learning classification tasks. The core of MSIMG is a density-based greedy peak selection algorithm, termed Density-Aware Peak Selection (DAPS), which dynamically identifies and extracts the most information-rich data regions. Figure 1 illustrates the complete workflow of MSIMG, from raw data processing to the final construction of the multi-channel image representation.

### 2.1. Mass Spec Matrix Construction

All raw LC-MS data are first converted into a universal mass spectrometry data format (e.g., mzML) and then transformed into a two-dimensional mass spec matrix through a unified preprocessing pipeline, which mainly consists of *m*/*z* binning and intensity normalization. Each LC-MS file contains a series of *m*/*z* values and their corresponding ion intensities at each retention time point. The purpose of *m*/*z* binning is to align the *m*/*z* axes across different samples and scan times. Specifically, given an *m*/*z* range $\left[mz_{\min},mz_{\max}\right]$ and a *bin size*, the index *i* for any *m*/*z* value is calculated as follows:(1)$$i=\lfloor \frac{mz-mz_{\min}}{bin\_size}\rfloor$$

All ion intensities falling into the same *bin* $B_{i}$ are summed to obtain the final intensity value $I_{B_{i}}$ for that bin:(2)$$I_{B_{i}}=\sum_{j\;s.t.\;\lfloor \frac{mz_{j}-mz_{\min}}{bin\_size}\rfloor =i}Intensity\left(mz_{j}\right)$$

After applying this operation to the mass spectra at all retention time points, each LC-MS file is converted into a two-dimensional mass spec matrix *M* with dimensions $\left(N_{scans},N_{bins}\right)$, where $N_{scans}$ is the number of retention time points and $N_{bins}$ is the total number of bins along the *m*/*z* axis.

To eliminate variations in overall signal intensity between samples and adapt the data to the input range of deep learning models, the constructed mass spec matrix *M* is normalized. Specifically, the 99.9th percentile of all non-zero elements in matrix *M* is set as the maximum value, and the minimum is set to 0. The normalized values in the matrix are then clipped to the range [0,1] to ensure numerical stability, yielding the final normalized mass spec matrix $M_{norm}$.

### 2.2. Density-Aware Peak Selection Strategy

The core of the MSIMG method lies in its data-driven strategy for generating candidate patch locations, the DAPS algorithm. This algorithm is designed to identify the most representative shared signal hotspot regions across the entire dataset and generate a set of candidate patch center coordinates. It discards the fixed-grid partitioning strategy in favor of dynamic, “information-density-centric” patch extraction.

The MSIMG method first processes each sample in the training set to generate a density map and extract a set of candidate peak points. The purpose of generating a density map is to identify potentially biologically significant signal-dense regions from the sparse and noisy mass spec matrix. This process aims to convert discrete signal points into a continuous density surface, thereby highlighting signal “hotspots” in the matrix that guide subsequent patch location selection. Specifically, an intensity threshold $\tau_{intensity}$ is set. Pixels in the normalized mass spec matrix $M_{norm}$ with intensity values greater than this threshold are considered effective signal points, while the rest are treated as background. This operation converts the continuous intensity matrix $M_{norm}$ into a binary signal mask matrix *S*. After obtaining the signal mask matrix *S*, a sliding window of size w×w is convolved over *S* to estimate the local signal density at each pixel, thus generating a density map *D*. Each pixel (i,j) in the density map *D* is the sum of all signal mask values within the w×w window centered at that point. The process can be described by the following equation:(3)$$D_{i,j}=\sum_{x,y\in {\mathrm{Window}}_{i,j}}S_{x,y}$$

The density map *D* is a 2D array of the same size as the mass spec matrix, where higher values indicate denser effective signals. However, bright regions (i.e., signal-dense areas) on the density map *D* may contain multiple adjacent local maxima. To precisely locate a single, most representative peak from each signal-dense region, MSIMG applies the Non-Maximum Suppression (NMS) algorithm. Originating from the field of object detection in computer vision, NMS is a key post-processing technique for eliminating redundant detection boxes [26]. Its core mechanism of selecting a single optimal target is well-suited for our peak localization task. In the MSIMG method, we adapt its core idea and implement peak selection through an iterative greedy strategy. Specifically, at the beginning of each iteration, the point with the global maximum density, $p_{max}$, is located in the current density map. If its density value is above a predefined density threshold $\tau_{density}$, it is recorded as a valid candidate peak. Simultaneously, to prevent re-selecting peaks from the same dense region in subsequent iterations, all density values within a rectangular area centered at $p_{max}$ (with the same size as the predefined patch size) are set to 0. This process continues until a preset number of candidate peaks are extracted or no points satisfying the density threshold can be found.

Through the above process, MSIMG extracts a set of high-quality, spatially separated candidate peak points from a single sample, which accurately represent the centers of the most signal-dense regions in that sample. This procedure is then repeated for every sample in the training set, and all extracted candidate peak points are aggregated to form a global set of candidate peaks, $P_{all}$. To select a final set of patch centers from this dense point set that best represents the dataset’s characteristics without spatial overlap, we designed a greedy selection algorithm—DAPS. This algorithm iteratively selects an “optimal peak” that maximizes local candidate peak coverage. To perform spatial queries efficiently, the global candidate peak set $P_{all}$ is first structured into a k-d tree [27]. In each iteration, DAPS evaluates the “potential value” of a subset of currently available candidate peaks. This value is defined as the number of candidate peaks contained within a circular region centered at a candidate peak $p_{i}$. This region is defined as a circle $B_{p,r}$ with a radius *r* equal to the distance from the patch center to its diagonal corner. Therefore, the potential value, ${\mathrm{Score}}_{p_{i}}$, of a candidate peak $p_{i}$ can be formalized as:(4)$${\mathrm{Score}}_{p_{i}}=\left|\left\{p_{j}|p_{j}\in P_{avail}\cap B_{p_{i},r}\right\}\right|$$
where $P_{avail}$ is the set of all currently available peaks and |·| denotes the number of elements in the set. DAPS then selects the peak with the highest score, $p_{best}$, as the optimal choice for the current iteration:(5)$$p_{best}=\underset{p_{i}\in P_{cand}}{\mathrm{argmax}}{\mathrm{Score}}_{p_{i}}$$

Once $p_{best}$ is selected, it is added to the final set of patch center coordinates, $P_{selected}$. Concurrently, all candidate peaks falling within its coverage area $B_{p_{best},r}$ are marked as unavailable to ensure that subsequent selections do not overlap. This iterative process continues until the density of the remaining available peaks falls below the preset threshold $\tau_{min\_peaks}$. DAPS ultimately outputs a set of candidate patch center coordinates, $P_{selected}$, that maximizes signal coverage across the entire training set.

### 2.3. Construction of the Multi-Channel Image Representation

After generating the final set of patch center coordinates, $P_{selected}$, using the DAPS algorithm, the next step is to construct the final multi-channel image representation for each sample. This is achieved through a unified, entropy-based scoring and ranking process based on these candidate locations. The process begins by extracting candidate patches for scoring. For each sample in the training set, MSIMG extracts patches from its corresponding normalized matrix $M_{norm}$ according to the candidate center coordinates in $P_{selected}$. Specifically, for each center coordinate (r,c) in $P_{selected}$, a patch of size patch_height × patch_width centered at that point is extracted. If the extraction area exceeds the matrix boundaries, it is padded with a preset value (e.g., 0). After this step, for each candidate location, we have the corresponding patch instances from all samples.

Next, MSIMG employs one-dimensional information entropy as the evaluation metric for the final patch selection process. For each candidate patch location, every sample in the training set has a corresponding patch. However, the amount of information contained in patches at the same location varies across different samples, resulting in different entropy values. Therefore, to select the locations that are most representative and information-rich across the entire training set, we first calculate the average information entropy for each candidate location across all training samples. The patch locations with the highest average information entropy are then chosen for the final selection. The entire workflow for this patch selection process is illustrated in detail in Figure 2. The core of this process is the calculation of information entropy. Specifically, for a given patch *P*, its information entropy H(P) is defined as:(6)$$H\left(P\right)=-\sum_{k=0}^{255}p_{k}\log_{2}\left(p_{k}\right)$$
where $p_{k}$ is the proportion of pixels with intensity value *k* (normalized to the 0–255 range) in patch *P*.

After calculating the average information entropy for all candidate patch locations, the top-ranking num_patches locations are selected as the “golden locations” for constructing the multi-channel image representation for all samples. Specifically, for every sample in the dataset (including both training and testing sets), MSIMG selects the patches at these num_patches golden locations and stacks them along the channel dimension. This forms a three-dimensional tensor with dimensions (num_patches, patch_height, patch_width). This 3D tensor is the “multi-channel image” representation of the sample and can be directly used as input for a CNN model, where num_patches corresponds to the number of input channels for the model.

### 2.4. Model Training and Evaluation Strategy

To ensure a robust and unbiased evaluation of the model’s performance, we designed a rigorous training and evaluation pipeline. As shown in Figure 3, the entire dataset is first divided into a training+validation set and an independent hold-out test set using an 8:2 stratified sampling split. On the training+validation set, we employed a 5-fold cross-validation strategy to train and select the model. For each fold, the model is trained on the training data and its performance is monitored on the validation data. All final performance evaluations are conducted on the hold-out test set to ensure the generalizability of the results.

To enhance model generalization and prevent overfitting, we applied data augmentation strategies exclusively to the training data. These included a series of random image transformations designed to simulate variations encountered during actual MS data acquisition, such as random affine transformations, random intensity scaling, addition of Gaussian noise, and random erasing of image regions. These augmentations ensure that the model learns more robust features that are insensitive to minor perturbations.

For model optimization, the Adam optimizer was used for efficient parameter updates. To prevent overfitting, L2 regularization (weight decay) was applied to penalize model complexity, helping the model learn more generalizable feature representations. To dynamically adjust the learning rate, we used a ReduceLROnPlateau scheduler: the learning rate was multiplied by 0.1 when the validation loss did not improve for 5 consecutive epochs. Furthermore, to address class imbalance in the dataset, we calculated class weights and incorporated them into the cross-entropy loss function. The entire training process was run for 64 epochs with a batch size of 8. We also implemented an early stopping mechanism: if the validation loss failed to improve for 10 consecutive epochs, training was terminated early, and the model weights with the best performance were saved. After completing the 5-fold cross-validation, we obtained five independently trained models. Each of these five models was then evaluated on the independent hold-out test set, and the mean and confidence interval of various performance metrics (such as accuracy, F1-score, etc.) were calculated and reported as the final performance of our method. All experiments were conducted using Python 3.9.23 and PyTorch 2.7.1 on a workstation equipped with an NVIDIA GeForce RTX 4070 Ti Super GPU.

### 2.5. Datasets

We conducted experiments on two independent, publicly available mass spectrometry datasets (SPNS and CD) to evaluate the effectiveness of our proposed MSIMG method. The SPNS dataset originates from a metabolomics and lipidomics study on Solitary Pulmonary Nodules (SPNs) [28]. The study found significant pattern differences in circulating metabolites and lipids between SPN patients (both benign and malignant) and healthy controls, but no significant differences between benign and malignant SPNs. Given this, and considering that distinguishing between benign and malignant nodules is beyond the scope of this study, we grouped the benign and malignant SPN samples into a single class. This created a binary classification task to distinguish SPN patients from healthy controls, comprising 880 SPN samples and 279 healthy control samples. The CD dataset is from a metabolomics study investigating the relationship between the host and gut microbiota in Crohn’s Disease (CD). This dataset includes a total of 294 fecal samples, with 250 from CD patients and 44 from healthy controls. These two datasets, representing different disease types and sample origins, provide a solid foundation for evaluating the performance and generalization capability of our method.

## 3. Results

This study compares the MSIMG method against two baseline methods from different paradigms. The first is the traditional peak list method, which represents the standard workflow in mass spectrometry analysis. This involves extracting feature peaks using common MS processing software, then converting the peak list into a one-dimensional feature vector for input into a classification model. The second baseline is the grid-based MetImage method, which uses a fixed-size sliding window to scan across the entire mass spec matrix at a fixed stride. This generates a corresponding set of candidate patches at fixed grid locations for each sample in the dataset. In our study, both the MSIMG and MetImage methods use the same subsequent process for patch selection and multi-channel image construction to ensure controlled experimental variables.

### 3.1. Parameter Sensitivity Analysis: Selection of Multi-Channel Image Dimensions and Number of Channels

In methods like MSIMG that represent mass spectrometry data as a multi-channel image, the number of selected patches, num_patches, directly determines the channel count of the final image and is a critical hyperparameter affecting model performance. To thoroughly investigate the impact of the number of input patches on model performance and to compare the feature selection efficiency of our proposed MSIMG(DAPS) against the baseline MetImage(GP), we conducted a series of parameter sensitivity experiments. Specifically, on both the SPNS and CD datasets, we systematically evaluated how the classification performance of downstream models was affected by an increasing number of input patches, using three different CNN backbone models (ResNet [29], DenseNet [30], and EfficientNet [31]). It is worth noting that the maximum number of candidate patches MSIMG can extract is determined by the signal density and distribution characteristics of the dataset itself; we considered these factors when setting the parameter range for our experiments.

The results of the patch number sensitivity analysis, as shown in Figure 4, reveal a clear and consistent trend: for both methods, increasing the number of patches generally leads to improved model performance across all tested conditions (with a fixed patch size of 224). This suggests that more patches provide richer information, helping the model learn more discriminative features. However, the two methods show significant differences in their performance improvement trajectories. Our proposed MSIMG method outperforms the MetImage method under almost all parameter settings and reaches its performance saturation point much earlier. For instance, on the SPNS dataset, regardless of the backbone model used, the DAPS strategy of MSIMG achieves optimal performance with an F1-score approaching or exceeding 0.95 using only 8 carefully selected patches. In contrast, the MetImage method still shows a considerable performance gap at 16 patches and requires many more patches to improve slowly. Furthermore, we observed that after MSIMG reaches its optimal performance, the F1-score curve flattens, with subsequent minor fluctuations being statistically insignificant and more likely due to experimental variance than a performance decline. This indicates that the DAPS strategy of MSIMG can efficiently identify the most representative regions in the dataset, thereby extracting key discriminative information.

In addition to the number of input patches, the patch size is another critical hyperparameter, as it determines the local information range the model can perceive in a single observation. To investigate the impact of this parameter, we systematically evaluated the effect of four different sizes on the classification performance of MSIMG(DAPS) and MetImage(GP), while keeping the number of patches fixed at 64. The results, presented in Figure 5, clearly demonstrate a non-linear relationship between patch size and model performance. A general observation is that excessively small patch sizes lead to a significant drop in performance. As the patch size increases, model performance tends to stabilize. This trend is particularly evident with the EfficientNetB0 model on the CD dataset, where performance for both methods jumps substantially when the patch size increases from 32 to 64, followed by a plateau as the size further increases from 64 to 224. This suggests that patches that are too small may fail to capture a complete, discriminative signal peak pattern, thus limiting the model’s learning capacity. Once the patch size is sufficient to contain key feature information, further increases do not yield significant performance gains. Considering performance stability and compatibility with mainstream CNN model inputs, we determined that 224 is a suitable size and used it in all subsequent comparative experiments.

This series of experiments leads to an important conclusion: both MetImage and MSIMG can be viewed as feature selection strategies, aiming to select the most discriminative subset of features (i.e., patches) from the high-dimensional mass spec matrix. The final classification performance is theoretically capped by the intrinsic information content of the dataset and the model architecture. The core difference between the two methods lies in the efficiency with which they approach this performance limit. The GP strategy, through a near-exhaustive approach, can eventually approach this limit by including a sufficient number of informative regions. In contrast, our proposed MSIMG method, with its data-driven, “density-peak-centric” intelligent selection strategy, can accurately locate and select these critical discriminative regions from the outset. This allows the DAPS strategy to reach the performance ceiling faster and more efficiently, with fewer patches than the GP strategy. Overall, the performance of the DAPS strategy not only demonstrates its effectiveness but also strongly validates that applying advanced, content-aware computer vision strategies to mass spectrometry data analysis is a promising and effective approach.

### 3.2. Performance Comparison of Multi-Channel Image Representation and Traditional Methods

To comprehensively evaluate the effectiveness of our proposed MSIMG multi-channel image representation method, we conducted a direct performance comparison with the traditional peak list method. Based on the preceding parameter sensitivity analysis, the MSIMG method was configured with a patch size of 224 × 224 and a patch number of 64. For experimental design, we equipped the traditional peak list method with a variety of classifiers. For the classic machine learning models, we utilized the implementations from the Python scikit-learn library to establish a performance baseline. Specifically, the Random Forest was used with its default hyperparameters, including 100 estimators. The Support Vector Machine was configured with a Radial Basis Function (RBF) kernel, while other parameters like regularization were kept at their default values. The Linear Discriminant Analysis was also used with its default settings. For all of these classical models, no further hyperparameter tuning was performed. In addition to these models, we also used three convolutional neural networks (ResNet50, DenseNet121, and EfficientNetB0) adapted for one-dimensional data. Correspondingly, for the two-dimensional multi-channel images generated using the MSIMG method, we used the standard 2D versions of the same three CNN models to ensure a fair and consistent comparison.

We conducted extensive comparative experiments on the SPNS and CD datasets, with the results presented in Table 1 and Table 2. The results clearly show that the MSIMG multi-channel image representation method comprehensively outperforms the traditional peak list method in terms of performance. On the larger SPNS dataset, this difference in performance is particularly stark. With the peak list method, both classic machine learning models and 1D deep learning models exhibited limited performance, with the highest F1-score reaching only 0.6671. In sharp contrast, the MSIMG method approached near-perfect performance across all three CNN models, with F1-scores consistently around 0.9977. This indicates that MSIMG provides a feature representation with extremely high information density and discriminative power, something that the one-dimensional feature vectors generated by traditional peak list methods struggle to achieve. A similar trend was observed on the smaller CD dataset. Although the peak list method performed reasonably well on this dataset, achieving a maximum F1-score of 0.7695, the MSIMG method still demonstrated a definitive advantage, with all three models surpassing an F1-score of 0.91, and the ResNet50 model reaching a high of 0.9402.

To further investigate the underlying reasons for the superior performance of our method, we conducted a comprehensive analysis of the intrinsic properties of the different data representations, with the results visualized in the Figure 6. The information content analysis reveals that the MSIMG (DAPS) representation possesses substantially greater Structural Richness and Dimensionality compared to both peak list methods, indicating that our approach captures a more complex and comprehensive feature space. Furthermore, we assessed the information redundancy within the 64-channel MSIMG representation. The low inter-channel correlation shown in the heatmap, combined with the fact that 57 principal components are required to explain 95% of the total variance, provides strong evidence that each channel contributes unique, non-redundant information. This highlights the efficiency of our DAPS selection strategy in creating an information-dense representation. Crucially, this superior data quality translates directly into classification power, as demonstrated by the comparative ROC analysis. The ResNet50 model trained on MSIMG data achieves a perfect Area Under the Curve (AUC) of 1.00, dramatically outperforming the models trained on the not-aligned (AUC = 0.73) and aligned (AUC = 0.60) peak lists. Collectively, this analysis confirms that the MSIMG framework provides a superior input paradigm for deep learning classifiers. Its success stems from a data-driven patch selection strategy that effectively captures and preserves the intrinsic structural and textural information within the data. By “encoding” this information into an image-based representation optimized for visual models, powerful CNNs can fully leverage their pattern recognition strengths, leading to breakthrough performance in mass spectrometry data analysis.

### 3.3. Visualization of the Effectiveness of Multi-Channel Image Representation

While the performance metrics in the preceding sections establish the superiority of the MSIMG method, we conducted further visualization analyses to intuitively understand the underlying reasons for this superior performance. This section aims to provide a direct visual comparison and evaluation from two key perspectives: the quality of the patches used as model inputs and the separability of the learned features that serve as model outputs. This will clarify how the data-driven strategy of the MSIMG method translates into an enhanced final classification performance.

To investigate the source of MSIMG’s feature advantage, we directly compared the visual quality of patches selected by our method with those selected by the baseline MetImage method. Figure 7 displays example patches (from channels 11 to 16) extracted by both methods on the SPNS and CD datasets, with a patch size of 224 × 224. There is a significant difference in the content of the patches generated by the two methods. Compared to MetImage’s fixed-grid partitioning strategy, the patches selected by MSIMG generally contain clear and continuous signal peak texture structures; particularly on the SPNS dataset, the extracted patches exhibit distinct clustered textures. In contrast, the content of patches selected by the MetImage strategy appears more random, often capturing either background noise with no discernible pattern or incomplete signal fragments that have been arbitrarily cut by the grid. For example, on the CD dataset, it is clear that its grid partitioning disrupts the integrity of complete mass spectral peak textures. These stark visual contrasts stem from MSIMG’s “density-peak-centric” strategy, which ensures that the selected regions are signal “hotspots” rather than random background areas. These high-quality, information-dense patches present clear visual patterns that reflect underlying, biologically meaningful molecular signals, providing the model with authentic biological patterns to learn from and forming the solid foundation for MSIMG’s superior performance.

An effective way to evaluate the quality of features extracted by different methods is to observe their distribution in a high-dimensional space. In this study, we used the t-SNE dimensionality reduction technique to compare the deep features represented by the 1D vectors from the traditional peak list method with those from the multi-channel images generated by the MSIMG method (after pooling and flattening). Figure 8 shows a t-SNE visualization comparison of the two methods on the SPNS and CD datasets. The subplots to the left of the dashed line clearly show that for the SPNS dataset, the feature vectors from the traditional peak list method result in severe mixing of sample points from different classes in the 2D space, making them nearly indistinguishable and indicating weak feature discriminability. In contrast, the features generated by the MSIMG method demonstrate vastly superior performance. As the number of input patches increases from 4 to 16, and then to 64, the class clusters in the t-SNE plots become progressively clearer and more compact. When 64 patches are used, the samples from different classes form almost completely separate and distinct clusters. This result provides strong evidence that the MSIMG multi-channel image representation can capture far richer and more precise discriminative information than traditional 1D vectors, thereby providing the downstream deep learning model with a high-quality, easily separable feature space.

## 4. Discussion

The experimental results of this study demonstrate that our proposed MSIMG method, by converting raw mass spectrometry data into a multi-channel image representation, achieves significantly superior classification performance compared to the traditional peak list method. This finding not only validates the effectiveness of the MSIMG method but, more importantly, prompts a re-examination of a fundamental issue in mass spectrometry data analysis: the relationship between data representation and the analytical objective

In traditional mass spectrometry data analysis, a peak list is typically generated for subsequent analytical tasks. The fundamental purpose of this process is to extract and interpret biologically meaningful information from complex, raw instrumental data [32]. A peak list is a structured, denoised, and information-compressed representation designed for human scientists. It discretizes continuous spectral data into a series of identifiable and annotatable metabolites or peptides for pathway analysis, with the ultimate goal of biological interpretation. However, when the analytical objective shifts from biological interpretation to a pure classification or prediction task, it is crucial to question whether this data representation, optimized for biological insight, remains the best choice for achieving optimal performance.

Deep learning models, particularly Convolutional Neural Networks (CNNs), derive their power not from understanding structured tables but from their exceptional ability to perceive spatial patterns, textures, and local correlations in images [33]. From this perspective, a traditional peak list represents a structurally mismatched input format for a CNN, with a significant loss of information. The success of MSIMG demonstrates that breakthrough performance can be achieved by redesigning the data representation to align with the demands of the task (i.e., classification) and the characteristics of the model. By converting mass spectrometry data into an image representation, MSIMG preserves the rich two-dimensional spatial information of the original data, providing the model with a more “native” and information-dense input, thereby achieving the classification goal more effectively.

This brings to mind the classic computer science principle of “Garbage In, Garbage Out.” Typically, this principle means that if the input data itself is of low quality, no model can produce reliable results [34]. However, our research reveals another facet of this principle: even information-rich raw data can become “garbage input” for a downstream model if it is inappropriately represented [35]. On the SPNS and CD datasets, the poor performance of the traditional peak list method was not due to low-quality raw mass spectrometry data, but because the one-dimensional vector representation produced after data filtering had lost a substantial amount of information, becoming an “information-poor” input. MSIMG provides a superior data representation strategy. It transforms high-quality raw mass spectrometry data into a high-quality model input through an appropriate representation method, thereby achieving satisfactory results. Notably, this data representation paradigm, optimized for Convolutional Neural Networks, may in turn offer new possibilities for enhancing the efficiency of extracting biological insights. Traditional peak list generation relies on fixed algorithms and parameters, which may overlook important low-abundance signals. In contrast, MSIMG employs an unbiased, data-driven process to precisely locate the most discriminative information-rich regions for the classification task. These patches, “attended to” by the model, are essentially signal hotspots that can guide researchers to focus on specific mass-to-charge ratio-retention time (*m*/*z*-RT) windows for more in-depth targeted analysis and identification. This strategy is expected to significantly enhance the efficiency of discovering potential biomarkers, thereby establishing an effective human-computer collaborative analysis loop.

## 5. Conclusions

This study introduces a novel framework named MSIMG, which transforms raw mass spectrometry data into multi-channel images for deep learning-based phenotype classification. By employing a data-driven, “peak-centric” patch selection strategy, MSIMG effectively preserves the integrity of key signal patterns, overcoming the critical limitations of traditional grid-based methods. Our extensive experiments demonstrate that the MSIMG method significantly improves classification performance across multiple datasets and deep learning architectures compared to traditional peak list methods. These results highlight the profound impact of data representation on model performance and validate the effectiveness of applying content-aware computer vision strategies to analytical chemistry data. This work paves the way for future research into more robust and interpretable deep learning models for clinical proteomics and metabolomics.

## Figures and Tables

**Figure 1 sensors-25-06363-f001:**
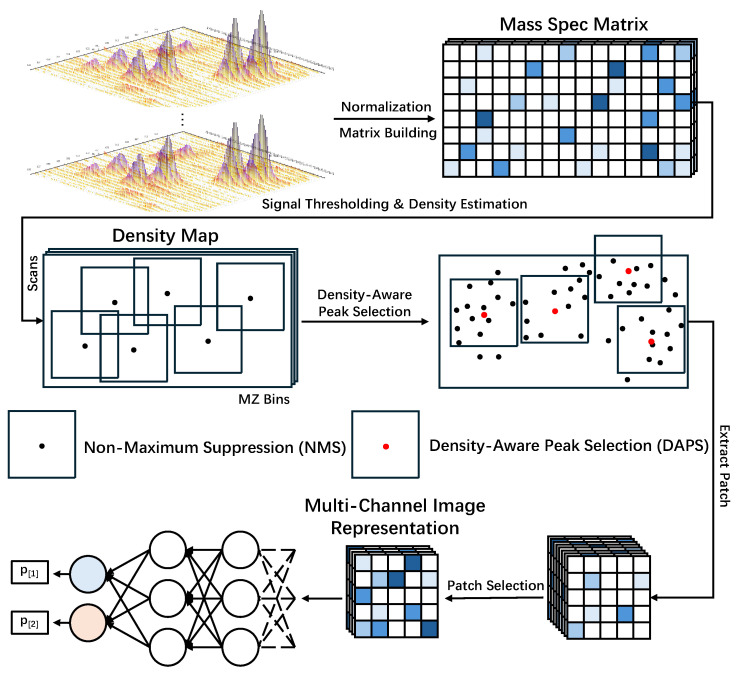
Schematic workflow of the MSIMG method for converting mass spectrometry data into a multi-channel image representation. The diagram illustrates how MSIMG transforms MS data into a multi-channel image. The process mainly includes: preprocessing raw data into a mass spec matrix; applying the DAPS algorithm (combining density map estimation and NMS) to determine candidate patch centers (represented by red dots), which are selected from a larger pool of initial peaks (black dots) identified across the dataset; and finally, selecting the most informative patches based on entropy scores and stacking them to form the final multi-channel image representation. Notably, the DAPS and final patch selection steps are performed on the entire training set to ensure the global representativeness of the features.

**Figure 2 sensors-25-06363-f002:**
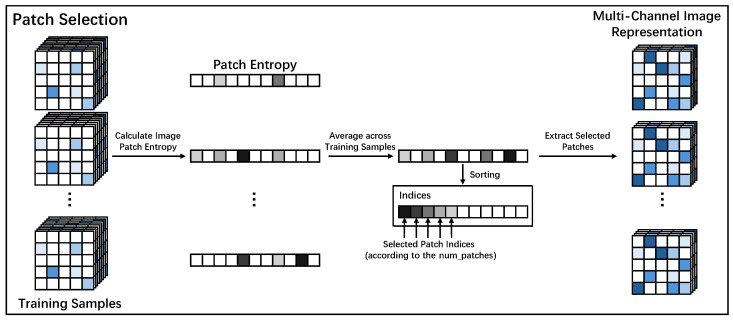
The Entropy-based patch selection workflow in MSIMG. For each candidate patch location, information entropy is calculated for the corresponding patch in every training sample. These values are then averaged across the training set to produce a single representative score for each location. The locations are ranked by their average entropy, and the top num_patches are selected. Patches from these top-ranked locations are extracted from every sample in the dataset and stacked to form the final multi-channel image representation.

**Figure 3 sensors-25-06363-f003:**
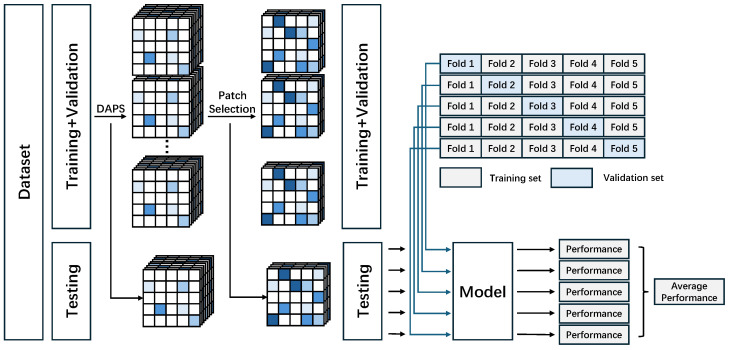
Model training and evaluation workflow of the MSIMG method. The dataset is first split into a training+validation set and an independent hold-out test set. The proposed DAPS algorithm and the subsequent patch selection process are executed only on the training+validation set to determine a set of the most informative patch locations. This selected set of locations is then applied to the test set to construct its corresponding multi-channel image representations, thus ensuring the independence of the test data.

**Figure 4 sensors-25-06363-f004:**
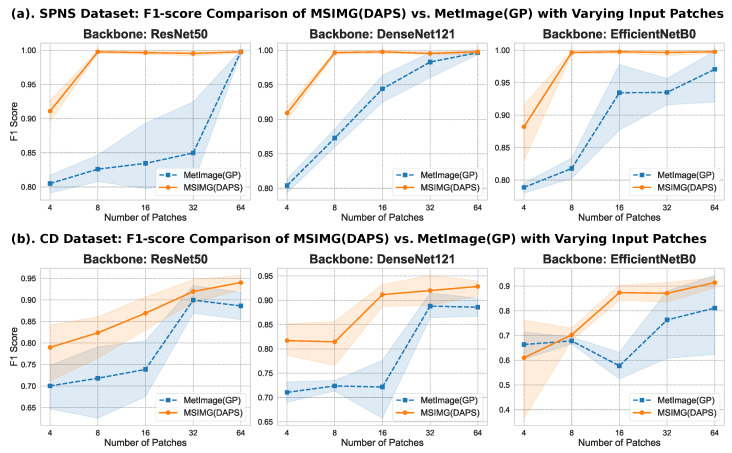
F1-score comparison between MSIMG(DAPS) and MetImage(GP) with varying numbers of input patches. This figure shows the impact of the number of selected patches on model performance with a fixed patch size of 224. (**a**) The performance trends on the SPNS dataset. (**b**) The performance trends on the CD dataset. For both datasets, the comparison was performed across three different CNN backbones (ResNet50, DenseNet121, and EfficientNetB0).The lines represent the mean F1-score across 5-fold cross-validation, and the shaded areas indicate the 95% confidence intervals.

**Figure 5 sensors-25-06363-f005:**
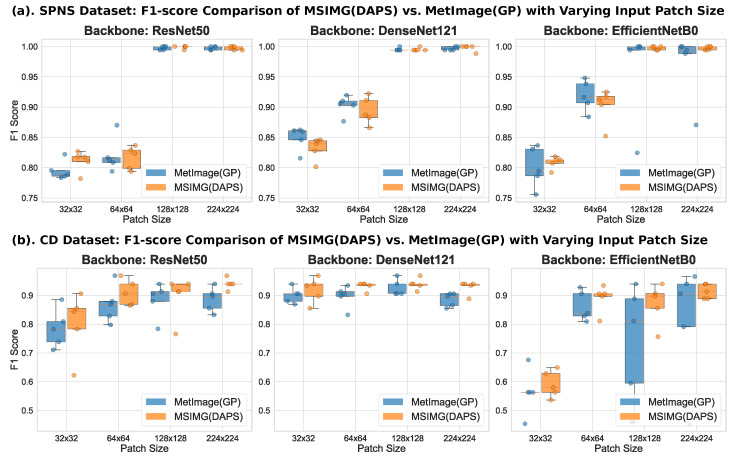
F1-score comparison between MSIMG(DAPS) and MetImage(GP) with varying input patch sizes. This figure shows the impact of patch size on model performance with a fixed number of 64 input patches. The performance comparison is shown for (**a**) the SPNS dataset and (**b**) the CD dataset. For both datasets, the experiments were conducted using three different convolutional neural network backbones (ResNet50, DenseNet121, and EfficientNetB0) across four different patch sizes. The box plots represent the quartile range of results from 5-fold cross-validation, with the center line indicating the median.

**Figure 6 sensors-25-06363-f006:**
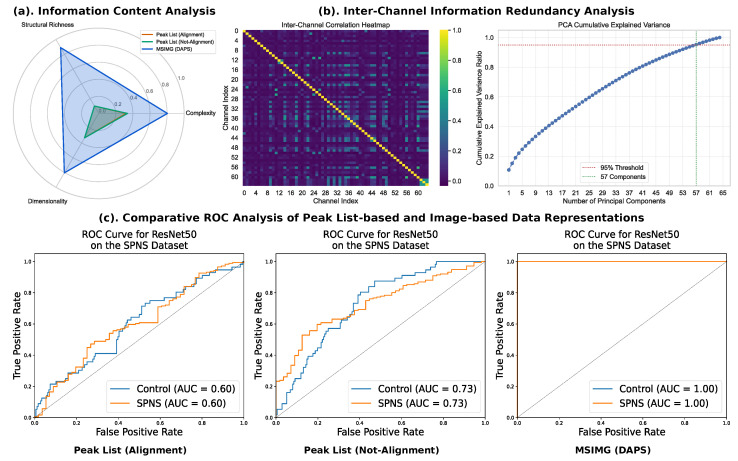
Comprehensive analysis linking data representation to classification performance. This figure evaluates three data preprocessing methods: Peak List (Alignment), Peak List (Not-Alignment), and our proposed MSIMG (DAPS). (**a**) Information Content Analysis via a radar chart comparing the three methods across normalized metrics of Complexity, Structural Richness, and Dimensionality. The MSIMG (DAPS) method exhibits substantially higher structural richness and dimensionality. (**b**) Inter-channel information redundancy analysis for the 64-channel MSIMG (DAPS) representation. The correlation heatmap (**left**) indicates minimal linear correlation between channels. The PCA cumulative explained variance curve (**right**), where the blue line plots the cumulative explained variance against the number of principal components, confirms low data redundancy, requiring 57 components to explain 95% of the total variance. (**c**) Comparative ROC curves for a ResNet50 classifier on the SPNS vs. Control classification task. The model trained on the MSIMG (DAPS) representation achieves a perfect Area Under the Curve (AUC) of 1.00, significantly outperforming models trained on the Peak List (Not-Alignment) (AUC = 0.73) and Peak List (Alignment) (AUC = 0.60) representations.

**Figure 7 sensors-25-06363-f007:**
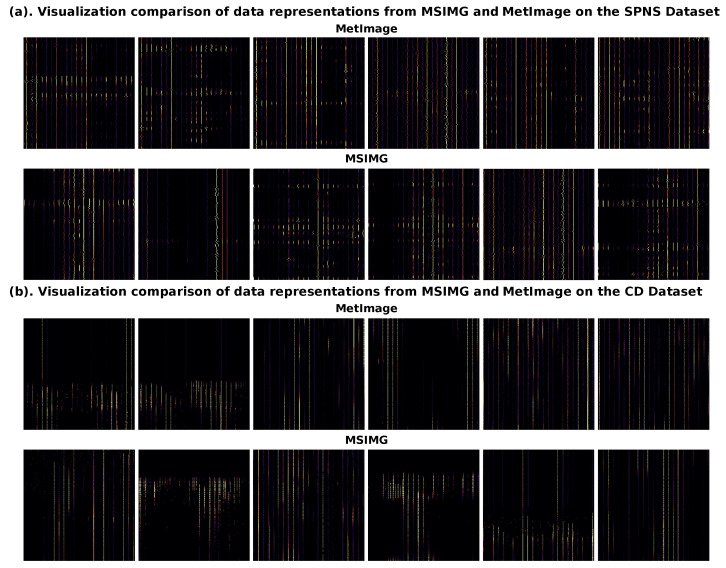
Visualization comparison of data representations from MSIMG and MetImage on the SPNS and CD dataset. The figure compares representative patches selected from (**a**) the SPNS dataset and (**b**) the CD dataset. For each dataset, the top row shows six feature patches (channels 11–16) generated by the grid-based MetImage method, while the bottom row shows the corresponding patches selected by the density-aware MSIMG method.

**Figure 8 sensors-25-06363-f008:**
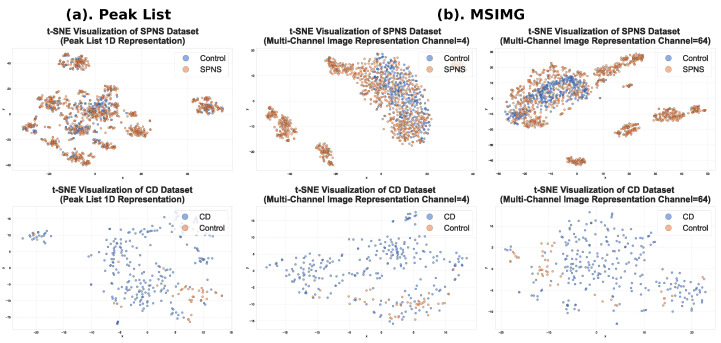
Comparison of the separability of features extracted by MSIMG and the traditional peak list method. This figure uses the t-SNE dimensionality reduction technique to visualize and compare the separability of features extracted by different methods. Subplot (**a**) shows the sample distribution based on the traditional Peak List 1D representation. Subplot (**b**) shows the sample distribution for the MSIMG multi-channel image representation, demonstrating improved class separation as the number of channels increases. The top and bottom rows show the results for the SPNS and CD datasets, respectively.

**Table 1 sensors-25-06363-t001:** Performance comparison of MSIMG and peak list methods on the SPNS dataset.

Model	Peak List (Align)		Peak List (Not-Align)		MSIMG	
	Accuracy	F1 Score	Accuracy	F1 Score	Accuracy	F1 Score
RF	0.7819 ± 0.01	0.5563 ± 0.02	0.8129 ± 0.01	0.6671 ± 0.02	-	-
SVM	0.7604 ± 0.00	0.4387 ± 0.02	0.7595 ± 0.00	0.4351 ± 0.01	-	-
LDA	0.5414 ± 0.02	0.4905 ± 0.02	0.5586 ± 0.03	0.5126 ± 0.03	-	-
ResNet50	0.7181 ± 0.05	0.4745 ± 0.05	0.6155 ± 0.03	0.5694 ± 0.02	0.9983 ± 0.00	0.9977 ± 0.00
DenseNet121	0.6991 ± 0.08	0.4699 ± 0.05	0.6500 ± 0.02	0.5966 ± 0.02	0.9983 ± 0.00	0.9977 ± 0.01
EfficientNetB0	0.7104 ± 0.06	0.4806 ± 0.05	0.6595 ± 0.04	0.5988 ± 0.03	0.9983 ± 0.00	0.9977 ± 0.00

**Table 2 sensors-25-06363-t002:** Performance comparison of MSIMG and peak list methods on the CD dataset.

Model	Peak List (Align)		Peak List (Not-Align)		MSIMG	
	Accuracy	F1 Score	Accuracy	F1 Score	Accuracy	F1 Score
RF	0.8542 ± 0.02	0.5689 ± 0.05	0.9085 ± 0.02	0.7695 ± 0.05	-	-
SVM	0.8440 ± 0.01	0.5298 ± 0.04	0.8780 ± 0.01	0.6424 ± 0.06	-	-
LDA	0.7593 ± 0.04	0.6760 ± 0.04	0.8542 ± 0.04	0.7580 ± 0.05	-	-
ResNet50	0.7424 ± 0.06	0.6238 ± 0.04	0.8542 ± 0.03	0.7677 ± 0.04	0.9661 ± 0.01	0.9402 ± 0.02
DenseNet121	0.7864 ± 0.16	0.6817 ± 0.16	0.8237 ± 0.09	0.7556 ± 0.10	0.9593 ± 0.02	0.9284 ± 0.02
EfficientNetB0	0.8339 ± 0.03	0.7035 ± 0.04	0.8237 ± 0.09	0.7355 ± 0.08	0.9492 ± 0.02	0.9138 ± 0.03

## Data Availability

The source code for MSIMG, along with all scripts used for data processing, statistical analysis, and data visualization in this study, is available on GitHub (https://github.com/NIM-NMDC/MSIMG). The public datasets used for the case studies in this research, SPNS and CD, were downloaded from the Metabolomics Workbench database under accession IDs ST001937 and ST003313, respectively.

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
