# Peer review of "MSIMG: A Density-Aware Multi-Channel Image Representation Method for Mass Spectrometry"

_sensors, 2025, doi:10.3390/s25206363_

Round 1
Reviewer 1 Report
Comments and Suggestions for Authors
The manuscript is well written and presents an innovative approach for application to mass spectrometry (MS) data. The authors compare their method with other algorithms from the literature and evaluate their figures of merit in order to demonstrate similar or distinct capabilities that the approaches can achieve.
However, among the approaches evaluated by the authors, all rely on feature selection; none employ widely recognized dimensionality reduction techniques such as PCA. Specifically, the authors use the traditional peak list method, a classic approach to extracting and selecting mass-to-charge peaks to create one-dimensional feature vectors. Another method used is the grid-based MetImage, which divides the spectral data matrix into a fixed grid creating uniformly sized patches (kernels), and the selection is done by heuristic measures such as entropy or mean value. The proposed MSIMG method applies an advanced signal density-based selection approach (DAPS).
Therefore, I am curious how an approach using classical PCA and its performance comparison would look like, similar to what was performed with the binned and kernel-based approaches demonstrated in the manuscript.
The authors chose an 80:20 split for the dataset; would there be differences using a 70:30 split? Does an 80:20 split not favor training by overfitting?
The manuscript uses the F1 score metric to assess model performance, which is appropriate and effective, especially in binary classification tasks with potential class imbalance, as is common in mass spectrometry data. The F1 score combines precision and recall into a single metric, offering a critical balance when false positives and false negatives carry different impacts.
Nevertheless, the ROC curve (Receiver Operating Characteristic) with AUC (Area Under the Curve) calculation is indeed another widely used metric that can provide complementary information regarding performance. The ROC curve evaluates the trade-off between true positive rate (sensitivity) and false positive rate across all threshold values, presenting a more comprehensive view of classifier behavior at different operating points.
I suggest applying the ROC curve because it would be a valuable informative complement to evaluate classifier performance across different thresholds. In clinical or biomedical contexts where false positives and false negatives have differing costs, ROC/AUC can offer a more complete picture of robustness and method utility.
One curiosity: did the authors consider using MSIMG as a preprocessing step before applying SVM, LDA, or other machine learning algorithms?
Author Response
Comment 1: I am curious how an approach using classical PCA and its performance comparison would look like, similar to what was performed with the binned and kernel-based approaches demonstrated in the manuscript.
Response 1: Thank you for this valuable suggestion. Following your recommendation, we have performed a new set of experiments applying PCA as a preprocessing step on the traditional peak list data before classification. The PCA model was fit exclusively on the training data to prevent information leakage and was configured to reduce the data to a fixed number of principal components. The results of these new experiments are shown below.
Performance Comparison of MSIMG and Peak List Methods on the SPNS Dataset
|
Model |
Peak List |
Peak List-PCA |
||
|
|
Accuracy |
F1 score |
Accuracy |
F1 Score |
|
RF |
0.8129±0.01 |
0.6671±0.02 |
0.7638±0.00 |
0.5368±0.01 |
|
SVM |
0.7595±0.00 |
0.4351±0.01 |
0.7638±0.01 |
0.4532±0.02 |
|
LDA |
0.5586±0.03 |
0.5126±0.03 |
0.7397±0.01 |
0.5822±0.01 |
|
ResNet50 |
0.6155±0.03 |
0.5694±0.02 |
0.6104±0.08 |
0.5504±0.04 |
|
DenseNet121 |
0.6500±0.02 |
0.5966±0.02 |
0.6241±0.05 |
0.5788±0.03 |
|
EfficientNetB0 |
0.6595±0.04 |
0.5988±0.03 |
0.7009±0.02 |
0.5796±0.06 |
Performance Comparison of MSIMG and Peak List Methods on the CD Dataset
|
Model |
Peak List |
Peak List-PCA |
||
|
|
Accuracy |
F1 score |
Accuracy |
F1 Score |
|
RF |
0.9085±0.02 |
0.7695±0.05 |
0.9017±0.01 |
0.7940±0.04 |
|
SVM |
0.8780±0.01 |
0.6424±0.06 |
0.8712±0.02 |
0.6471±0.03 |
|
LDA |
0.8542±0.04 |
0.7580±0.05 |
0.8814±0.02 |
0.6847±0.07 |
|
ResNet50 |
0.8542±0.03 |
0.7677±0.04 |
0.8203±0.07 |
0.7017±0.08 |
|
DenseNet121 |
0.8237±0.09 |
0.7556±0.10 |
- |
- |
|
EfficientNetB0 |
0.8237±0.09 |
0.7355±0.08 |
0.8915±0.02 |
0.7342±0.05 |
The results of this comparison were nuanced. On the SPNS dataset, we observed that PCA preprocessing modestly improved the performance for models like LDA and SVM. However, for Random Forest and the deep learning models, the performance either slightly decreased or remained stagnant. Similarly, on the CD dataset, PCA offered a slight benefit to the Random Forest and SVM models but led to a decrease in performance for LDA and the deep learning models. Given the lack of a consistent performance improvement from PCA across the different models and datasets, we have chosen not to include these specific results in the final manuscript tables to maintain focus on our core comparisons. We have presented the full results here for your reference and hope you understand our reasoning.
Most importantly, we found that even in the cases where PCA improved the performance of the peak list method, the results are still significantly lower than those achieved by our proposed MSIMG method. For example, the highest F1-score for the Peak List-PCA approach on the SPNS dataset was 0.5988, whereas the MSIMG method consistently achieved F1-scores around 0.9977. This finding strongly reinforces our central conclusion: the data representation paradigm is the critical factor. The 2D image-based representation of MSIMG, when paired with a CNN, is fundamentally more effective than a 1D feature vector approach.
Finally, regarding the application of PCA to our kernel-based MSIMG method for classification, we did not perform this experiment for a critical methodological reason. The core purpose of MSIMG is to preserve the 2D spatial relationships and textural patterns within the mass spectrometry data by creating an image-like representation. Applying PCA would require "flattening" these 2D patches into 1D vectors, which would destroy the very structural information our method is designed to capture and leverage. Such an operation would be contrary to the fundamental hypothesis of our work. However, inspired by your suggestion, we did find an excellent application for PCA in analyzing the properties of our MSIMG-generated data. We used it to assess the inter-channel information redundancy, confirming the high information density of our representation. This new analysis has been added to Section 3.2. We sincerely thank you for this insightful comment, as it has led to an important addition that further strengthens our manuscript.
Comment 2: The authors chose an 80:20 split for the dataset; would there be differences using a 70:30 split? Does an 80:20 split not favor training by overfitting?
Response 2: Thank you for this important question regarding the dataset split and the potential for overfitting. This is a critical aspect of model validation, and we appreciate the opportunity to clarify our methodology.
Our choice of an 80:20 split for the training/validation and hold-out test sets is a standard and widely adopted practice in the machine learning field. This ratio is generally considered to provide a good balance, ensuring that the model has a sufficiently large and diverse dataset to learn from, while still reserving a substantial, independent portion of the data for a robust final evaluation.
More importantly, we have implemented a comprehensive strategy specifically to mitigate the risk of overfitting, ensuring that the model's performance is generalizable and not just an artifact of the training data. As detailed in our manuscript, this strategy includes several key components:
- Independent Hold-Out Test Set: The 20% test set was strictly used as a hold-out set. All final performance evaluations were conducted exclusively on this unseen data, which ensures an unbiased assessment of the model's generalization capabilities.
- 5-Fold Cross-Validation: On the 80% portion of the data, we employed a 5-fold cross-validation strategy for model training and selection. This ensures that our model is robust and its performance is not dependent on a single, specific validation split.
- Data Augmentation: To enhance generalization, we applied various data augmentation techniques, such as random affine transformations and the addition of Gaussian noise, exclusively to the training data.
- Regularization: We incorporated L2 regularization (weight decay) to penalize model complexity and an early stopping mechanism that terminates training if the validation loss does not improve, both of which are powerful techniques to prevent overfitting.
Regarding a 70:30 split, while it is also a valid approach, we are confident that our conclusions would remain consistent. Given the rigorous anti-overfitting measures we have in place, the model learns generalizable features rather than memorizing the training set. Therefore, a modest change in the split ratio would be unlikely to alter the central finding of our paper—that the MSIMG representation method significantly outperforms traditional approaches. While the exact performance metrics might show minor fluctuations, the overall trend and the demonstrated superiority of our method would hold.
Comment 3: I suggest applying the ROC curve because it would be a valuable informative complement to evaluate classifier performance across different thresholds.
Response 3: Thank you for this excellent suggestion. We completely agree that the ROC curve and the associated Area Under the Curve (AUC) provide a more comprehensive and robust evaluation of a classifier's performance, especially for the kind of binary classification task presented in our study.
Following your recommendation, we have performed this analysis and included a new figure in the manuscript to directly compare the different data representation methods using ROC curves. This new analysis can be found in Section 3.2, specifically in "Fig 6. Comprehensive analysis linking data representation to classification performance."
As shown in subplot (c) of that figure, the ROC analysis clearly demonstrates that the model trained on our MSIMG representation achieves a perfect AUC of 1.00, significantly outperforming the models trained on the peak list-based methods.
We appreciate you bringing this up, as this addition has provided a stronger, more direct visualization of our method's superior performance.
Comment 4: Did the authors consider using MSIMG as a preprocessing step before applying SVM, LDA, or other machine learning algorithms?
Response 4: That is an excellent and insightful question. Thank you for this constructive suggestion.
Our primary motivation for developing MSIMG was to create a data representation paradigm specifically tailored to the strengths of Convolutional Neural Networks (CNNs). The central hypothesis of our paper is that by preserving the two-dimensional spatial and textural information of the original mass spectrometry data within image-like patches, we can enable CNNs to leverage their powerful ability to automatically learn hierarchical patterns.
Using MSIMG as a preprocessing step for models like SVM or LDA would require flattening our multi-channel image tensor into a one-dimensional feature vector. This action would, by design, discard the crucial intra-patch spatial relationships that our method aims to preserve and that CNNs are built to interpret. While the resulting vector might still be information-rich, this approach would likely underutilize the core advantage of the MSIMG representation. Such flattening operations would destroy the structural information of the extracted data, which is contrary to the fundamental purpose of our method. Therefore, we did not perform this part of the experiment, but we still agree that this is an insightful suggestion.
Our experiments comparing the end-to-end MSIMG+CNN framework against traditional models (like SVM and LDA) using standard peak list features already demonstrate the significant performance gains of our image-based paradigm (as shown in Tables 1 and 2).
Reviewer 2 Report
Comments and Suggestions for Authors This manuscript proposed a MSIMG framework to the selection strategy that capture the core features of phenotypes. MSIMG is based on a density-based greedy peak selection algorithm, and is featured on selection on a set of LCMS data instead of individual data. It seems this technique is attractive and have potential for all MS-realted tasks. Nevertheless, some of the issues should be improved: 1. The software and hardware should be reported for this work. 2. MetImage is a method used for comparison for SPNs dataset. Why it is not used in dataset 2, instead, peak list is used for classification? 3. The peak list method, as for high-precision metabolomics, is also missing detail. For a fair comparison, peak list should be refined by peak alignment due to the chromatographic and mass spectrometric drift. Since the authors did not provide peaklist detail, I assume no processing is applied, which is not a valid comparison. 4. The code and data availability should also be addressed. The CD dataset lacks reference source. 5. The training detail for all the algorithms in Table 1 is missing. Hyperparameter tuning process is also not mentioned, even for random forest and svm. Please add such details.Author Response
Comment 1: The software and hardware should be reported for this work.
Response 1: Thank you for pointing out this omission. We agree that this information is important for reproducibility. As requested, we have now added a sentence summarizing the hardware and software environment at the end of the "Model Training and Evaluation Strategy" section. To further aid reproducibility, a more comprehensive list of all software packages and their specific versions will also be provided in our public GitHub repository.
Comment 2: Why is MetImage not used in dataset 2, instead, peak list is used for classification?
Response 2: Thank you for your question and for the opportunity to clarify our experimental design. We apologize if the structure of our results section caused any confusion.
A direct comparison between our proposed MSIMG method and the grid-based MetImage method was indeed performed on both the SPNS and the CD datasets. The results of this comparison are presented in the "Parameter Sensitivity Analysis" section, specifically in Figure 4 and Figure 5. In both of these figures, the top row of plots displays the performance on the SPNS dataset, while the bottom row shows the performance on the CD dataset. These figures illustrate the performance (F1-score) of both MSIMG and MetImage on the CD dataset across a range of hyperparameters, including varying numbers of patches and different patch sizes. This analysis demonstrated that our MSIMG method consistently outperforms the MetImage approach.
The subsequent section, "Performance Comparison of Multi-Channel Image Representation and Traditional Methods," and its corresponding Table 2, was designed for a different purpose. Having established the superiority of MSIMG over MetImage, we then aimed to compare our optimized imaging-based paradigm against the fundamentally different, traditional peak list method. Therefore, in Table 2, we present the results of MSIMG versus the peak list method on the CD dataset to highlight the advantages of our data representation strategy over conventional workflows.
In summary, we chose this two-stage comparison to first validate our specific patch selection strategy against a similar imaging method (MetImage) and then to demonstrate the broader advantage of our method over the traditional analysis paradigm (peak list).
Comment 3: For a fair comparison, peak list should be refined by peak alignment due to the chromatographic and mass spectrometric drift. Since the authors did not provide peaklist detail, I assume no processing is applied, which is not a valid comparison.
Response 3: Thank you for this critical and insightful comment. You are correct in your assumption that our initial experiments on the peak list method did not include a peak alignment step to correct for chromatographic and mass spectrometric drift. We agree that this is a crucial step for a fair and robust comparison.
To address this, we have now incorporated a peak alignment process into our peak list extraction workflow. We then re-ran all the relevant classification experiments using this new, aligned data. The updated results have been added to Table 1 and Table 2 under a new column, "Peak List-Align", for a direct comparison against the unaligned peak list and our MSIMG method.
The results were surprising at first, yet upon reflection, they are also quite logical. Across both the SPNS and CD datasets, the performance of the classifiers on the aligned peak list data (Peak List-Align) was consistently and significantly lower than the performance on the original, unaligned Peak List data.
While this outcome may seem counterintuitive, as peak alignment is a standard and important practice, we hypothesize that for these specific datasets, the alignment process may have inadvertently reduced the information density of the final feature vectors. It is possible that the alignment algorithm, in its effort to standardize peak locations, subtly distorted or averaged out some of the fine-grained, discriminative patterns that the models were able to capture from the unaligned data.
Based on this new finding, we have now performed a deeper comparison of how the information content of different data representations impacts final performance. We have created a new comparative figure and added a corresponding analysis to the manuscript to illustrate this relationship. This new content can be found in Section 3.2. We sincerely thank you for your insightful comment, as it has directly led to this new analysis and has significantly strengthened our manuscript.
Comment 4: The code and data availability should also be addressed. The CD dataset lacks reference source.
Response 4: Thank you for your valuable feedback.
We would like to clarify that we have included a "Data Availability Statement" section following the Conclusions to address the availability of our code and data. In this section, we state that the source code for MSIMG is publicly available on GitHub, and we provide the specific accession IDs for both the SPNS (ST001937) and CD (ST003313) datasets from the Metabolomics Workbench database.
Regarding the reference for the CD dataset, we appreciate you pointing this out. We conducted a thorough search for a publication associated with this dataset from the original providers but were unfortunately unable to find one. However, to ensure transparency and reproducibility, we have explicitly provided the dataset's accession number (ST003313). This allows any researcher to access the complete dataset and its associated metadata directly from the public repository.
Comment 5: The training detail for all the algorithms in Table 1 is missing. Hyperparameter tuning process is also not mentioned, even for random forest and svm.
Response 5: Thank you for your valuable feedback regarding the training details.
We would like to clarify that the training details for our deep learning models (ResNet, DenseNet, and EfficientNet), including the data augmentation strategies, optimizer (Adam), learning rate scheduler (ReduceLROnPlateau), loss function (cross-entropy with class weights), and early stopping mechanism, have been described in detail in Section 2.4, "Model Training and Evaluation Strategy".
We acknowledge that the corresponding details for the traditional machine learning models were not included. To address this, we have now expanded Section 3.2 to include the implementation details for the Random Forest (RF), Support Vector Machine (SVM), and Linear Discriminant Analysis (LDA) models. For these comparative experiments, we utilized the standard implementations from the scikit-learn library. To establish a baseline performance, we used their default hyperparameters and did not perform an exhaustive hyperparameter tuning process. We have now made this explicit in the manuscript. We appreciate you bringing this to our attention, as it has improved the completeness of our methods description.
Reviewer 3 Report
Comments and Suggestions for Authors
Interpreting the vast amount of data obtained from MS spectra, particularly from the perspective of proteomics or metabolomics, is an extremely tedious task, carrying with it the possibility of misinterpretation or ‘loss’ of important data. Hence, the development of a universal model that allows for quick interpretation of the results obtained is extremely important. The presented research is part of a trend that uses deep learning to build a universal model that allows for quick and accurate interpretation of the obtained data.
Below are a few suggestions regarding the research described:
Line 52 – overfitting model – in what sense? Shouldn't it also be added that such a model cannot be generalised, which reduces its universality?
Figure F5 – is not very clear
Does Figure 6 compare exactly the same data? I have the impression that the interpretation of this figure in the text is more elaborate than the figure itself. Because it is definitely less legible.
Figure 7 – the font size is inappropriate. The descriptions in the figures can only be read when greatly enlarged. Perhaps it would be worth considering rearranging the individual diagrams to make them more legible.
Author Response
Comment 1: Line 52 – overfitting model – in what sense? Shouldn't it also be added that such a model cannot be generalised, which reduces its universality?
Response 1: Thank you for this insightful suggestion. You are correct that simply stating "overfitting" is not specific enough. Overfitting in this context refers to the model learning the noise and specific artifacts of the training set rather than the generalizable biological signals, due to the excessive number of parameters relative to the sample size. This indeed leads to a model that cannot be generalized to new data, reducing its universality.
We have revised the manuscript to clarify this point, explicitly stating that an overfitted model in this scenario would fail to generalize to new, unseen samples, thereby limiting its utility.
Comment 2: Figure F5 – is not very clear.
Response 2: Thank you for your feedback regarding Figure 5. We have now completely revised the figure to improve its legibility. In the new version, we have significantly enlarged the font size for all labels and added more detailed descriptions to ensure the content is clear and easy to understand. We hope that the revised figure now meets your expectations.
Comment 3: Does Figure 6 compare exactly the same data? I have the impression that the interpretation of this figure in the text is more elaborate than the figure itself. Because it is definitely less legible.
Response 3: Thank you for your careful observation and feedback on Figure 6. We appreciate the opportunity to clarify its content and purpose.
To answer your first question, yes, the figure compares patches derived from the exact same data file. The top row displays patches selected by the MetImage method, while the bottom row shows patches selected by our MSIMG method from that single sample, specifically showing the content from channels 11 to 16.
We understand your point regarding the figure's legibility and why our textual interpretation may seem more elaborate than what is immediately visible. The reason for the figure's appearance is a direct result of the inherent nature of mass spectrometry data, which is extremely sparse. When a small 224x224 region of the data is visualized, the vast majority of pixels correspond to background with zero intensity, while only a few pixels contain the actual signal peaks. This sparse appearance is normal for this type of data at this scale.
This is precisely why our textual interpretation is more detailed. The crucial difference, which is subtle visually but significant for the model, is what these sparse signals represent. Our MSIMG method is designed to center the patches on these signal "hotspots," thereby capturing meaningful and complete peak structures. In contrast, the grid-based MetImage method often selects patches that contain only background noise or arbitrarily cut fragments of a peak. This fundamental difference in the quality of the selected information is what we aimed to elaborate upon in the text.
We hope this clarifies the comparison and the reason for the figure's appearance. We appreciate your understanding of the challenges involved in visualizing such sparse data.
Comment 4: Figure 7 – the font size is inappropriate. The descriptions in the figures can only be read when greatly enlarged. Perhaps it would be worth considering rearranging the individual diagrams to make them more legible.
Response 4: Thank you for your feedback on the legibility of our figure. We have thoroughly revised this figure to address your concerns. Please note that due to the addition of another figure in the manuscript, this figure has now been renumbered to Figure 8. In the new version, we have significantly increased the font size of all text and labels. Following your suggestion, we have also rearranged the individual diagrams to make them larger and clearer, which we believe greatly improves the overall readability.
We hope the revised Figure 8 is now clear and effectively addresses your concerns.
Round 2
Reviewer 2 Report
Comments and Suggestions for Authors
The response and proper revisions were made in this manuscript according to my advice, and I think it is ready to move on to the publication phase.